# Exploring the Relationship between Physical Activities and Health-Related Factors in the Health-Related Quality of Life among People with Disability in Korea

**DOI:** 10.3390/ijerph19137839

**Published:** 2022-06-26

**Authors:** Taeeung Kim, So-Youn Park, In-Hwan Oh

**Affiliations:** 1Department of Preventive Medicine, School of Medicine, Kyung Hee University, Seoul 02447, Korea; ktang7711@gmail.com; 2Department of Medical Education and Humanities, School of Medicine, Kyung Hee University, Seoul 02447, Korea; ukii77@gmail.com

**Keywords:** health-related quality of life, disabled people, physical activity, health-related factor, Korea

## Abstract

The purpose of this study is to explore the relationship between modes (e.g., frequency and total time) of physical activity and health-related conditions of disabled people on their health-related quality of life (HRQoL) in Korea. This study is a cross-sectional research funded by the Ministry of Health and Welfare. Data was obtained from the 2017 disability survey. A total of 6549 people with disabilities (Mage = 61.92, SD = 17.36; Male = 55.98%) were analyzed in this study. The higher the frequency of physical activity for the disabled in Korea, the more positive the HRQoL (*p* < 0.001). Among the elderly disabled, the higher the severity of disability and educational degree, the lower the HRQoL (all *p* < 0.05). Disabled people who had fewer diseases and lived an independent socio-economic and cultural life had a higher HRQoL (all *p* < 0.001). This study revealed different dimensions of how health-related factors influence the quality of life of people with disabilities. More attention should be paid to supporting people in being independent and active, in order to help them maintain a healthy life. Especially, the barriers to physical activity faced by disabled people are multi-layered and multifaceted. Increasing the frequency of physical activity for disabled people is not only beneficial for their physical function, but also for their HRQoL. This study enables welfare promotion for disabled people through various policies and incentives. Further, this will be an opportunity to reduce the socio-economic burden on medical and health-related services related to the disabled population.

## 1. Introduction

The number of disabled people registered with the Ministry of Health and Welfare of Korea is currently 2,633,026 (5.1%), due to aging, traffic accidents, and industrial disasters [1]. This number has more than doubled from 1.45 million (3.09% of the population) in 2000 to 2.63 million (5.1% of the population) in 2020 [1,2]. In particular, among the 15 types of disabilities, the proportion of people with developmental disabilities such as intellectual disabilities and autism, increased from 7.0% in 2010 to 9.4% in 2020 [1].

Disabled adults have significant difficulties in walking, climbing, vision, hearing, memory, concentration, or decision-making. They are three times more likely to develop heart disease, or to be affected by a stroke, diabetes or cancer than adults without disabilities [3]. Aerobic physical activities or exercises can help reduce the negative effects of these chronic conditions, but nearly half of disabled adults do not engage in these during their spare time [3]. This lack of physical activity among disabled people causes a greater risk of chronic diseases (e.g., obesity, hypertension, diabetes, etc.) than in non-disabled people [3]. 

Disabled people have significantly worse health-related conditions when compared to the general population, including poor overall health, lack of adequate health services, drinking, smoking, and inactivity [3]. However, they do have preventable health-related conditions such as obesity, mental health, depression, pain, and fatigue [3]. Therefore, it is possible that they pursue the same well-being and happiness as the general population, by preventing and modifying negative health-related conditions. 

Health-related quality of life (HRQoL) is a widely pursued concept. It is the modern expansion of health and well-being defined as a state of complete physical, mental and social well-being, and not simply a state without disease or weakness [4]. Health and medical research on HRQoL has been actively conducted in various fields (e.g., dementia, diabetes, obesity, mental health, cancer, oral, eye, HIV/AIDS, sexuality, exercise, etc.) [5,6,7,8,9,10,11,12,13,14] and in multilayered fields with various respondents and topics (e.g., older adults, homeless, adolescents, veterans, pregnant women, people with disabilities, healthcare, etc.) [6,8,12,15,16,17].

According to previous studies, HRQoL is known to be affected by factors such as health behavior (drinking and smoking) [18,19], exercise [20,21,22], job status [23,24,25], independent living [26,27,28], and chronic disease [29,30,31,32]. These studies emphasized quality of life (QoL) or well-being that encompasses the body and mind. Therefore, through personal health management, QoL can be improved and the socio-economic burden can be reduced.

Studies on HRQoL for disabled people included: exploring factors that affect the lives of patients with specific diseases (e.g., strokes, schizophrenia, psoriasis, musculoskeletal pain, heart failure, multiple sclerosis, etc.) [33,34,35,36,37], older adults [38], specific patients (e.g., multiple sclerosis and chronic ankle symptom) and people with disabilities [39,40,41,42,43]; or exploring the extent of changes in HRQoL brought about by exercises or other interventions [38,44,45]. Studies on the HRQoL of disabled people, based on health-related factors (e.g., health behaviors such as drinking and smoking, degree of chronic disease, exercise participation, and health-related entities that affect HRQoL) were insufficient or lacking. Therefore, this study aims to explore the relationship between modes (e.g., frequency and total time) of physical activity and health-related conditions of disabled people in Korea. The hypotheses of our study are as follows:HRQoL of people with disabilities has an association with health-related conditions.HRQoL of people with disabilities has an association with exercise frequency.HRQoL of people with disabilities has an association with amount of exercise.

## 2. Materials and Methods

### 2.1. Study Sample and Design

#### Data Source

This study is a cross-sectional research, with data obtained from the 2017 disability survey. It was funded by the Ministry of Health and Welfare [46]. Regular surveys were conducted with disabled people to understand their prevalence in the Korean population, their living conditions and their welfare needs. Surveyors visited families with disabled members and conducted structured interviews. Data such as relationship with each household member, gender, age, the disability registration status, enrollment year, registered disability type, disability rating, household size, household income and expenditure, major income source, home-ownership/rent, etc. were collected. 

Further, with respect to the legally defined diverse disabilities (e.g., physical disability, brain lesion, visual impairment, hearing impairment, speech impairment, intellectual disability, autism, mental impairment, renal dysfunction, etc.), data on the nature of the disability (date of occurrence, cause, etc.), health-medical conditions and needs, daily living support, assistive devices, education, employment and work life, socio-cultural and recreational activities, marital status, life satisfaction and experiences of violence and/or discrimination, housing, welfare services, and financial status were collected. A total of 36,200 of the 44,161 families in the 250 surveyed regions responded in a valid manner. The sample had 6549 people with disabilities, who were analyzed in this study. The relevant Institutional Review Board approved this study (case number: 1040198-210923-HR-146-01).

## 3. Measurement Instruments

### 3.1. Dependent Variable (EQ-5D-5L)

The HRQoL scale used in this study was EQ-5D-5L. It consists of five domains (mobility, self-care, usual activities, pain/discomfort, and anxiety/depression), with each domain being composed of a five-point scale (1 = no problem, 2 = slight problems, 3 = moderate problems, 4 = serious problems, and 5 = extreme problems). The health status of the EQ-5D-5L is defined by taking one level in each of the five areas; 11111 represents the best and 55555 represents the worst. The formula for calculating the utility weight of Korea (N4 model) was presented in a previous study [47]. The EQ-5D-5L utility index is calculated by assigning weights according to the grade of each domain. A utility score of 1.0 on each domain scale means perfect health, zero equals death, and possible utility values range from −0.066 to 1. 

### 3.2. Independent Variables

The purpose of this study is to explore the relationship between modes of physical activity and health-related factors in HRQoL of disabled people in Korea. Based on previous studies, the following independent variables related to HRQoL were utilized. First, the independent variables that correspond to the socio-economic background of the disabled individual were considered: age, gender, educational degree, income status, marital status, and disability grades. The age of the participants in this study ranged from 0 to 99 years. The educational degree was initially collected from an eight-point Likert scale (1 = preschool to 8 = graduate schools) and was revised into four categories as follows: (1) “elementary school or less,” (2) “middle schools,” (3) “high schools,” and (4) “universities or more.” Monthly income was initially measured as a continuous variable, on a five-point Likert scale modified as follows: (1) “less than 1 million won,” (2) “more than 1 million won and less than 2 million won,” (3) “more than 2 million won and less than 3 million won,” (4) “more than 3 million won and less than 4 million won,” and (5) “more than 4 million won.” Prior to this categorization, the income range in Korean won was converted to US dollars using the average exchange rate of the Bank of Korea in 2017 [48] as follows: (1) “less than $885,” (2) “more than 885$ and less than $1769,” (3) “more than $1769 and less $2654,” (4) “More than $2654 and less than $3538,” and (5) “More than $3538.” Marital status was initially measured on a six-point scale (1 = married to 6 = other). For the purpose of this study, it was modified into a binary variable of marital status (1 = married or 0 = others: widowed, divorced, separated, or single). The disability grade was measured on a six-point Likert scale according to the Korean Health and Welfare State 2018-151, with the implication that the higher the grade number, the less severe the disability [49]. 

Independent variables include smoking, drinking alcohol, work activity status, being able to go out alone, degree of discomfort when outside home, difficulty using transportation, number of chronic diseases, and frequency and total time (minutes) of physical activity. Smoking status was initially measured on a four-point scale (1 = never smoked and still does not smoke, 4 = daily smoke). It was modified as a binary variable of smoking status. Alcohol consumption was determined by the number of drinks on a six-point scale (1 = never drank in the last year to 6 = 4 or more per week), and average alcohol consumption on a five-point scale (1 = not drank at all in the last 1 year, to 6 = 10 or more drinks). Composite variables were used (lowest 1 to highest 36).

On a 16-point scale, the question of the previous week’s activity (1 = worked, to 16 = other) was modified with a binary variable to determine whether people were active at work. The capacity of venturing out alone was measured as a binary variable. The degree of discomfort during out-of-home activities was specified using a four-point scale (from 1 = not at all uncomfortable to 4 = very uncomfortable). The initial number of chronic diseases was specified on a 25-point scale (1 = high blood pressure, to 25 = other diseases). For this study it was modified to a four-point scale (0 = 0, to 3 = more than 3 chronic diseases). The degree of difficulty in using transportation was measured on a four-point scale (1 = not at all difficult, to 4 = very difficult). Finally, physical activity frequency was measured on a seven-point scale (1 = not currently exercising, to 7 = almost daily) and physical activity was assessed by summing the number of minutes spent on each exercise. The types of physical activity in this study are as follows: daily activities (e.g., breathing exercise, walking, jogging, bare hand gymnastics or stretching, balancing exercise, weight training), organized physical activity (e.g., climbing, swimming, badminton,), classes (e.g., yoga, water walking, water gymnastics) out-of-home activities (e.g., bicycle, gateball, table tennis, bowling, billiards, and park golf or golf).

### 3.3. Statistical Analysis

All data analyses were conducted using STATA version 15.1 (College Station, TX, USA). Descriptive statistics (e.g., *t*-tests, independent *t*-tests) and one-way ANOVA were used to report participants’ characteristics and variable correlations, respectively. Multiple linear regression was performed to capture the relationship between physical activity and HRQoL-controlling health-related conditions among the respondents. For the linear regression analyses, normality, linearity and homoscedasticity of residuals’ assumptions were checked, which appeared to be accurate for the analyses. [50]. To check for multivariate outliers and multicollinearity, regression with outlier statistics and collinearity diagnostic were employed and all assumptions were achieved (VIF value < 4.0, and tolerance < 0.2) [50]. Finally, all independent variables showed less than 5% of missing entries, which was not a statistically significant difference between groups (e.g., missing vs. not-missing) of any variable, and hence, it was ignored.

## 4. Results

### 4.1. Descriptive Statistics

As shown in Table 1, a total of 6549 disabled people participated in the research analysis of the relationship between the frequency of physical activity and health-related factors on HRQoL of disabled people. The average age was 61.92 years (deviation = 17.56), with 3666 (about 56%) comprising men. The disability grades varied, but grade 5 (1513 people; 23.7%) and grade 6 (1554 people; 24.3%) accounted for about 50%. Results related to income showed about 72% of participants earning less than 4 million won per month. About 87% attained high school education or less. 

### 4.2. Main Results

This study aimed to explore the HRQoL and health factors of disabled people that relate to the frequencies of physical activity. As seen in Table 2, our main research hypothesis suggests that the frequency of physical activity has a positive impact on the HRQoL. Compared with the disabled people who did not exercise in the previous year, the higher the frequency of exercise, the higher the statistical significance of HRQoL. The effect size also increased with twice a week (Coe. = 0.0218; 95% CI = 0.0074, 0.0362), three or more times a week (Coe. = 0.0277; 95% CI = 0.0163, 0.0392), and almost every day (Coe. = 0.0324; 95% CI = 0.0227, 0.0421). However, the amount of exercise did not have a statistically significant relationship with HRQoL.

It was identified that as age increased by one unit, the HRQoL decreased by 0.0004 (Coe. = −0.0004; 95% CI = −0.0007, −0.0001). As for the HRQoL on the disability grade, the less severe the disability, the higher the HRQoL as follows: Grade 2 (Coe. = 0.0695; 95% CI = 0.0509, 0.0880), Grade 3 (Coe. = 0.0611; 95% CI = 0.0428, 0.0793), Grade 4 (Coe. = 0.0670; 95% CI = 0.0480, 0.0860), Grade 5 (Coe. = 0.0723; 95% CI = 0.0537, 0.0909), and Grade 6 (Coe. = 0.0738; 95% CI = 0.0552, 0.0925). The educational degree also appeared to have a statistically significant effect on the HRQoL. The higher the educational level, the higher the negative impact on the HRQoL, as follows: middle school (Coe. = −0.0180; 95% CI = −0.0278, −0.0082), and higher than college (Coe. = −0.0223; 95% CI = −0.0342, −0.0105).

During data collection, the HRQoL of disabled people who had worked in the previous week was predicted to be higher than those who did not (Coe. = 0.0220; 95% CI = 0.0142, 0.0299). Additionally, it was found that disabled people who ventured out independently had a very positive effect on HRQoL compared to those who did not (Coe. = 0.1205; 95% CI = 0.1080, 0.1331). An increase in the idea that out-of-home activity is inconvenient predicted increasing negative HRQoL (Coe. = −0.0543; 95% CI = −0.0597, −0.0491). Higher difficulty in transportation usage predicted higher negative factors of HRQoL (Coe. = −0.0366; 95% CI = −0.0418, −0.0313). Finally, it was confirmed that HRQoL worsened as the number of chronic diseases increased, compared to people without them: 1 (Coe. = −0.0231; 95% CI = −0.0336, −0.0126), 2 (Coe. = −0.0527; 95% CI = −0.0636, −0.0418), and 3 or more (Coe. = −0.0797; 95% CI = −0.0902, −0.0692). 

## 5. Discussion

The purpose of this study was to explore the relationship between the modes (e.g., frequency and total time) of physical activity and health-related conditions of disabled people on their HRQoL in Korea. This study served as an opportunity to provide specific information about the effect of physical activity on the HRQoL of disabled people. It was confirmed that providing more frequent physical for disabled people is effective in improving their HRQoL. People with disabilities have higher barriers to performing physical activity (e.g., program restrictions, economic issues, emotional and psychological barriers, limited information of places and spaces, lack of social support, limited information on facilities and programs, and lack of fitness and healthcare professionals) [51]. 

The study of physical activity has positive effects on HRQoL, which has been well-organized for adolescents [7], the elderly [20,22,38], people with disabilities, and disease-affected patients [21,32]. In this study, the QoL of disabled people in Korea is closely related to the QoL of people who engage in physical activities, as well as their overall QoL. Furthermore, the physical activity of disabled people contributes to improving their HRQoL. Therefore, disabled people can experience psycho-social healing and stability, since physical activities enable exchange with guardians and other disabled people who participate in the programs.

This study shows that increasing the frequency of physical activity (e.g., breathing exercise, walking, jogging, bare hand gymnastics or stretching, balancing exercise, weight training climbing, swimming, badminton, yoga, water walking, water gymnastics, bicycle, gateball, table tennis, bowling, billiards, and park golf or golf), rather than the amount of exercise, positively affects HRQoL of disabled people. Most of the present research is concerned with the effect of physical activity intervention programs on QoL [7], e.g., a six-month intervention for the psychological and mental health of the elderly, and QoL through physical activity participation [20], QoL and clinical motor symptoms through a five-week exercise treatment program according to disability grade [21], a six-month home-based exercise intervention for the elderly in Switzerland regarding falls, QoL and exercise adherence [22], association between the cost of chronic disease in primary care in Brazil, and QoL, physical activity, and drug use [32], and the impact of long-term follow-up (2.6 years) on physical activity and performance on QoL [38]. Although existing studies depict exercise interventions as having a positive effect on QoL, specific information on specific modes of physical activity (e.g., frequency, total amount of time) were absent. Therefore, this study contributes to the literature by stating that the frequency of exercise plays a significantly more important role than the overall amount of exercise on the HRQoL of disabled people. By overcoming different sets of barriers and limitations, disabled people can continue to continue exercising for longer and more frequently, ultimately benefiting their physical and mental health. 

This study found that QoL of disabled people in Korea deteriorated with age. This is consistent with a cross-sectional study of factors related to disability and QoL among the elderly in the Polish community [42]. A physiological phenomenon deteriorates the QoL of disabled people. The effect of disability severity ratings on QoL is consistent with other results [33,52]. Policymaking and implementation of social welfare policies for the disabled in Korea should be undertaken carefully according to the severity of the disability, considering that the greater the severity of a disability, the lower the HRQoL.

Access to society for the disabled in Korea is very limited [2]. Therefore, this study showcases the trend of higher education resulting in lower QoL for the disabled in Korea. Resultantly, even if disabled people study harder for a better job or social status, social inequality limits their access to society, resulting in lesser income, lower social status, and a sense of despair or frustration that affects their QoL.

The workplace activities of disabled people in Korea were found to have a positive effect on HRQoL. This is supported by research on work status and QoL in patients with a disease (e.g., cancer, kidney transplant) and adolescents [23,24,25]. Disabled people have several socio-economic and physical limitations to engaging in stable regular jobs [3]. However, as shown in this study, for disabled people who engaged in work activities, this had a significantly positive impact on their HRQoL. Therefore, it is important that welfare policymakers and implementers not only find a solution to their difficulties (e.g., physical movement and pain), but also help them achieve socio-economic independence to improve their QoL in the long term. 

Factors related to the independence of disabled people in Korea (e.g., going out alone, inconvenience in outside activities, and difficulty in transportation) were found to be highly related to the HRQoL of the disabled, which corresponds with other research [26,27,28]. Therefore, disabled people who venture out alone without the help of family, guardians and acquaintances, have no difficulty using transportation, and have little inconvenience performing outside activities, are predicted to have significantly positive HRQoL by increasing efficacy and confidence. Therefore, to improve the QoL of the disabled, socio-cultural independence must be considered along with socio-economic independence. 

Finally, it was confirmed that disabled people with a higher number of chronic diseases had an increasingly negative HRQoL. This can be confirmed using various studies (e.g., elderly, disabled, adolescents) [29,30,32]. This can be attributed to the fact that an increase in the number of diseases results in an increase in socio-economic burden, a weak and low functioning body, and depression and feelings of insecurity. Therefore, to improve the HRQoL of the disabled, quality medical and health policies are extremely important.

This study should be interpreted in light of the following limitations. Firstly, this study utilized a multiple-item and self-reported measurement to assess all study variables from the respondents’ one-day recollection. Therefore, the result of this study might not be accurate in its entirety. Additionally, respondents may overestimate their physical activity levels and other variables due to societal bias. Next, this study is a cross-sectional analysis of the relationship between physical activity, health-related status, and HRQoL of disabled people in Korea. Therefore, the results of this study should not be interpreted as a causal relationship. In the future, intervention experiments are needed to establish a causal relationship between the QoL of the disabled and the exercise mode (e.g., frequency, intensity, total time). Lastly, since this study focuses on Korean disabled people, it is limited in its generalization and interpretation with respect to disabled people from different socio-cultural backgrounds. Nevertheless, this study was able to provide specific information on the effect of the mode of exercise on the HRQoL of the disabled in Korea. Hence, it is expected to serve as an important reference material when formulating policies. 

## 6. Conclusions

Disabled people need healthcare and wellness programs for the same reasons as non-disabled people. Although the HRQoL of the disabled cannot be dramatically improved with physical activity and health-related factors alone, several studies have verified that the QoL of the disabled has a positive effect on their personal and public life. Current research provides additional evidence related to the HRQoL of people with disabilities and indicates that QoL outcomes for people with disabilities can be influenced by both individual socioeconomic and health-related factors (e.g., being independent, being more active, chronic disease). This study revealed different dimensions of how health-related factors influence the quality of life of people with disabilities. More attention should be paid to supporting people in being independent and active to help them maintain a healthy life. Confirmation that various and multi-layered health-related factors are influencing the quality of life of the disabled could bring about a systematic and deep understanding of the HRQoL of people with disabilities, due to the complexity of the HRQoL of people with disabilities. In addition, the barriers to physical activity related to the disabled are multilayered and multifaceted. In this study especially, increasing the frequency of physical activity of the disabled is not only beneficial for their physical function, but also positively affects their HRQoL. This study will enable welfare promotion for disabled people through construction and implementation of various policies and incentives. Therefore, a long-term goal of this study is to improve the understanding of individual and environmental connections with the mechanisms of HRQoL of people with disabilities, which can lead to more effective public health prevention and intervention strategies that increase the HRQoL of people with disabilities. Furthermore, this presents an opportunity to reduce the socio-economic burden on medical and health-related services concerned with disabled people.

## Figures and Tables

**Table 1 ijerph-19-07839-t001:** Characteristics of study participants.

Variables	N (Mean)	% (SD)	*p*-Value
Age	6549 (61.92)	100 (17.36)	<0.001
Gender			<0.001
Male	3666	55.98	
Female	2883	44.02	
Disability levels (higher number = lower level of severity)			<0.001
1	370	5.78	
2	840	13.13	
3	1123	17.56	
4	997	15.59	
5	1513	23.65	
6	1554	24.29	
Income levels			<0.001
Less than $885	1706	26.05	
More than $885 less than $1769	1864	28.46	
More than $1769 less than $2654	1109	16.93	
More than $2654 less than $3538	833	12.72	
More than $3538	1037	15.83	
Education attainment			<0.001
Less than elementary	2883	44.15	
Middle school	1051	16.09	
High School	1767	27.06	
Greater than college	829	12.70	
Smoking	6470 (1.71)	99% (1.06)	<0.001
Amount of drinking	6470 (5.89)	99% (8.52)	<0.001
Working last week			<0.001
Yes	2360	36.66	
No	4077	63.34	
Going out independently			<0.001
Yes	5590	85.36	
No	959	14.64	
Inconvenience of out-of-home activities	6286 (2.45)	96 (0.89)	<0.001
Number of chronic diseases			<0.001
0	1286	19.64	
1	1351	20.63	
2	1387	21.18	
Greater than 3	2525	38.56	
Difficulty in using transportation	6286 (2.30)	96 (0.90)	
Marital status			<0.001
Yes	3609	56.54	
No (widowed, divorced, separated, or single)	2774	43.46	
Physical activity frequency			<0.001
Not currently exercising	2648	40.43	
Less than once a month, less than 10 times a year	41	0.63	
1–2 times a month	96	1.47	
Once a week	221	3.37	
Twice a week	444	6.78	
3 or more times a week	942	14.38	
Almost everyday	2157	32.94	
Amount of physical activity (minutes)	6549 (31.74)	100 (39.25)	<0.001
Total Number of Participants	6549		

Notes. N: Number; SD: standard deviation; data source: the 2017 disability survey.

**Table 2 ijerph-19-07839-t002:** Multivariate linear regression of physical activity and health-related conditions on quality of life for people with disabilities.

Dependent Variable	Quality of Life (EQ-5D-5L)		Coe.	95% CI	*p*-Value
Independent Variables	Age		−0.0004	−0.0007, −0.0001	<0.05
	Gender (Male)		0.0067	−0.0017, 0.0150	0.12
	Disability levels				
	1		-	-	-
	2		0.0695	0.0509, 0.0880	<0.001
	3		0.0611	0.0428, 0.0793	<0.001
	4		0.0670	0.0480, 0.0860	<0.001
	5		0.0723	0.0537, 0.0909	<0.001
	6		0.0738	0.0552, 0.0925	<0.001
	Income levels				
	Less than $885		-	-	-
	More than $885 less than $1769		0.0049	−0.0043, 0.0142	0.30
	More than $1769 less than $2654		0.0039	−0.0070, 0.0148	0.49
	More than $2654 less than $3538		−0.0017	−0.0139, 0.0105	0.79
	More than $3538		0032	−0.0085, 0.0149	0.59
	Education attainment				
	Less than elementary		-	-	-
	Middle school		−0.0180	−0.0278, −0.0082	<0.001
	High School		−0.0067	−0.0161, 0.0026	0.16
	Greater than college		−0.0223	−0.0342, −0.0105	<0.001
	Smoking		−0.0036	−0.0072, 0.0001	0.06
	Amount of drinking		−0.0002	−0.0006, 0.0002	0.36
	Working last week		0.0220	0.0142, 0.0299	<0.001
	Going out independently		0.1205	0.1080, 0.1331	<0.001
	Inconvenience of out-of-home activities		−0.0543	−0.0597, −0.0491	<0.001
	Number of chronic diseases				
	0		-	-	-
	1		−0.0231	−0.0336, −0.0126	<0.001
	2		−0.0527	−0.0636, −0.0418	<0.001
	Greater than 3		−0.0797	−0.0902, −0.0692	<0.001
	Difficulty in using transportation		−0.0366	−0.0418, −0.0313	<0.001
	Marital status		−0.0034	−0.0107, 0.0040	0.37
	Physical activity frequency				
	Not currently exercising		-	-	-
	Less than once a month, less than 10 times a year		−0.0021	−0.0438, 0.0396	0.92
	1–2 times a month		0.0176	−0.0104, 0.0455	0.22
	Once a week		0.0094	−0.0104, 0.0291	0.35
	Twice a week		0.0218	0.0074, 0.0362	<0.05
	3 or more times a week		0.0277	0.0163, 0.0392	<0.001
	Almost everyday		0.0324	0.0227, 0.0421	<0.001
	Amount of physical activity		0.0001	−0.0001, 0.0002	0.57
	Total number of participants			6549	

Notes: Coe: coefficient; CI: confident interval; data source: the 2017 disability survey.

## Data Availability

The data used in this study are available at https://data.kihasa.re.kr/kihasa/kor/contents/ContentsList.html (accessed on 10 November 2021).

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
