# Peer review of "Exploring the Relationship between Physical Activities and Health-Related Factors in the Health-Related Quality of Life among People with Disability in Korea"

_ijerph, 2022, doi:10.3390/ijerph19137839_

Round 1

Reviewer 1 Report

In general, I consider the work interesting and carefully written. My comments concern the methodological part and conclusions.

In my opinion “Increasing physical activity ”( Line 18 and line 299) is not an appropriate wording as it is underlined in the paper that: “ This study shows that the number of physical activities, rather than the amount of 227 exercise, positively affects HRQoL of the disabled.”

In the methodological part and in the discussion, it should be described in more detail what type of physical activity was taken into account: daily activity, workplace activities, in the form of organized physical activity like exercises, classes, out-of-home activities. The health-related factors should also be precisely listed and discussed.

The conclusions are not correctly formulated, as most of them are not based on the results obtained in the presented research.

Line 186- “heal-related” instead “health-related”

Author Response

Dear reviewer,

Thank you for reviewing our paper. Your valuable comments were very important in making our paper better. Thank you for your valuable and sharp comments. We have revised the following according to your comments and advice.

In general, I consider the work interesting and carefully written. My comments concern the methodological part and conclusions.

In my opinion “Increasing physical activity ”( Line 18 and line 299) is not an appropriate wording as it is underlined in the paper that: “ This study shows that the number of physical activities, rather than the amount of 227 exercise, positively affects HRQoL of the disabled.”

=> As you pointed, we revised “Increasing physical activity ” to “. Increasing frequency of physical activity” in the line of 20 and 323 with red

In the methodological part and in the discussion, it should be described in more detail what type of physical activity was taken into account: daily activity, workplace activities, in the form of organized physical activity like exercises, classes, out-of-home activities. The health-related factors should also be precisely listed and discussed.

=> The types of physical activity in this study are as follows in the methodical part with the red line 154- 160;.daily activities (e.g., breathing exercise, walking, jogging, bare hand gymnastics or stretching, balancing exercise, weight training), organized physical activity like exercises (e.g., climbing, swimming, badminton,), classes (e.g., yoga, water walking, water gymnastics) out-of-home activities (e.g., bicycle, gateball, table tennis, bowling, billiards, and park golf or golf)

=> The type of physical activity was stated in the discussion section with red line 238 to 241.

“activities (e.g., breathing exercise, walking, jogging, bare hand gymnastics or stretching, balancing exercise, weight training climbing, swimming, badminton, yoga, water walking, water gymnastics, bicycle, gateball, table tennis, bowling, billiards, and park golf or golf)”

The conclusions are not correctly formulated, as most of them are not based on the results obtained in the presented research.

=> The following conclusions were added in the line of 312 to 322 with red

 “Current research provides additional evidence related to the HRQoL of people with disabilities and Current research provides additional evidence related to the HRQoL of people with disabilities and indicates that  QoL outcomes for people with disabilities can be influenced by both individual socioeconomic and health-related factors (e.g., being independent, being more active, chronic disease). This study revealed different dimensions of how health-related factors influence the quality of life of people with disabilities. More attention should be paid to provide being independent and active to help them maintain a healthy life. Confirming that various and multi-layered health-related factors are influencing the quality of life of the disabled, it could bring about the understanding of the HRQoL of people with disabilities systematically and deeply due to the complexity of the HRQoL of people with disabilities..”

=> The following conclusions were added in the line of 326 to 329 with red

“Therefore, a long-term goal of this study is to improve the understanding of individual and environmental connections with the HRQoL of people with disabilities mechanisms, which can lead to more effective public health prevention and intervention strategies to increase the HRQoL of people with disabilities.”

Line 186- “heal-related” instead “health-related”

=> We corrected it to health-related” in the line 196 with red

In-Hwan Oh, M.D., Ph. D

Department of Preventive medicine

School of Medicine

26 Kyungheedae-ro, Dongdaemun-gu, Seoul,

Republic of Korea, 02447

Tel: 82-2-961-2304, Fax: 82-2-969-0792

E-mail: parenchyme@gmail.com

Reviewer 2 Report

The purpose of this study was to explore the relationship between the modes of physical activity and health-related conditions of the disabled on their HRQoL in Korea.

The research results are very interesting and important, but also quite obvious. The Authors have presented their results thoroughly and reliably.

Here are some questions and suggestions:

- Data were collected in 2017. Why did the Authors wait so long to publish the results?

- How was the recruitment for research carried out?

- In my opinion, it would be advantageous to have a chart showing the number of people at each stage of the study and the reason for exclusion from the studies.

Author Response

Thank you for reviewing our paper. Your valuable comments were very important in making our paper better. Thank you for your valuable and sharp comments. We have revised the following according to your comments and advice.

The purpose of this study was to explore the relationship between the modes of physical activity and health-related conditions of the disabled on their HRQoL in Korea.

The research results are very interesting and important, but also quite obvious. The Authors have presented their results thoroughly and reliably.

Here are some questions and suggestions:

- Data were collected in 2017. Why did the Authors wait so long to publish the results?

=> The year 2017 used in this study is the most recent data in data analysis. The Korean Ministry of Health and Welfare collects data every three years, but the latest 2020 data are not yet available.

- How was the recruitment for research carried out?

=> The household identification survey and in-depth interview survey were conducted by the Korean Ministry of Health and Welfare. Household visit interview survey by investigators based on a structured questionnaire. Survey Population: As of 2017, general residents living in general residential facilities in 17 cities and provinces nationwidehousehold and member of the household. Sample size: About 45,000 households in 250 sampling districts (average 180 households per survey district)

- In my opinion, it would be advantageous to have a chart showing the number of people at each stage of the study and the reason for exclusion from the studies.

=> Among the total number of households, only people with disabilities collected this data, and no disabled participants were excluded for analysis.

In-Hwan Oh, M.D., Ph. D

Department of Preventive medicine

School of Medicine

26 Kyungheedae-ro, Dongdaemun-gu, Seoul,

Republic of Korea, 02447

Tel: 82-2-961-2304, Fax: 82-2-969-0792

E-mail: parenchyme@gmail.com

Reviewer 3 Report

Thank you for the opportunity to review this study, which reports cross-sectional associations between physical activity and health-related conditions of disabled people on their health-related quality of life.  The target group of the study is very important, and the results add new information to be utilized in future studies.

The study is based on a large sample of disabled individuals with a large age range. The text is well-written and easy to follow. The introduction provides rational for the study and the purpose of the study is clearly presented.

I found few points that might be useful to revise or to clarify further:

-          Title: Since the study is based on cross-sectional data the use of the term “effect on” may be misleading. Please consider revising.

-          Abstract: The conclusions seem not to arise from the results presented. Please adjust.

-          Introduction, lines 47-49: Health-related quality of life has been defined using the reference [4]. Please check the relevance of the reference for this purpose.

-          Materials and methods, lines 95-95: It is stated that 36,200 families responded in a valid matter. However, only 6549 persons were analyzed in this study. Please give more details about these figures. For example: How was the study sample drawn? How did you select the people included into the analyses?

-          Measurement instruments, line 112: Please remove the brackets around QoL or revise the sentence otherwise.

-          Measurement instruments, line 117: The age range of the participants was wide (0-99), the mean age being 62 years. Please give some more details about the distribution of age and the data collection. How was data collected among the youngest participants? The questions described on lines 117-152 seem to be more feasible among the adults. How about the informed consent? How was it collected from the children?

-          Table 1: It is not clear, why there is the title Quality of life in the table. It seems that the results (N, mean, %, SD) are presented according to row titles. Please specify or revise.

-          Table 1: The categories of marital status (yes, no) are not clear either from the table or from the lines 128-129. Please specify this.

-          Table 1: Please indicate the unit for the amount of physical activity. On the line 152 it is stated that physical activity was assessed according to the number of minutes devoted to exercise every time. If 6549 stands for the minutes per each exercise, it sounds very high. If it covers total activity time, a time frame would be needed. Furthermore, since the variables presented in the table 1 are categorical, it might be more informative to present just the N and percentage.

-          Table 2: Please write the DV and IV with the full words. I suppose they mark for dependent variable and independent variables, but it would be clearer to give exact words.

-          Table 2: Please indicate the adjustments in the footnote.

-          Results: Did you consider analyzing potential interactions?

-          Discussion, line 210: It is stated that the purpose was to explore the relationship between the modes of physical activity and health-related conditions of disabled people on their health-related quality of life. However, based on the results, modes of physical activity were not analyzed, but the frequency and amount instead. Please adjust.

-          Discussion, lines 213-215: I agree with this sentence, but I did not find support for it from the results. I encourage you to replace “opportunities for physical activity” with “more frequent physical activity”.  

-          Discussion, line 227: I would encourage you to write “frequency of physical activity” (meaning number of physical activity sessions) instead of “number of activities”.

-          Discussion, line 241: In addition to “longer time” you could point out also “more frequently”.

-          Please be specific when writing HRQoL and when QoL. There seems to be some inconsistency throughout the text now.

Author Response

Thank you for reviewing our paper. Your valuable comments were very important in making our paper better. Thank you for your valuable and sharp comments. We have revised the following according to your comments and advice.

Thank you for the opportunity to review this study, which reports cross-sectional associations between physical activity and health-related conditions of disabled people on their health-related quality of life.  The target group of the study is very important, and the results add new information to be utilized in future studies.

The study is based on a large sample of disabled individuals with a large age range. The text is well-written and easy to follow. The introduction provides rational for the study and the purpose of the study is clearly presented.

I found few points that might be useful to revise or to clarify further:

-Title: Since the study is based on cross-sectional data the use of the term “effect on” may be misleading. Please consider revising.

=> As you mentioned, we revised it to “Exploring the relationship between Physical Activities and Health-Related Factors in the Health-Related Quality of Life among People with Disability in Korea”

-Abstract: The conclusions seem not to arise from the results presented. Please adjust.

=> We added more statement in the abstract for conclusion  in the line of 17 to 19 such as “This study revealed different dimensions of how health-related factors influence the quality of life of people with disabilities. More attention should be paid to provide being independent and active to help them maintain a healthy life.”

-Introduction, lines 47-49: Health-related quality of life has been defined using the reference [4]. Please check the relevance of the reference for this purpose.

=> We checked and corrected it.

-Materials and methods, lines 95-95: It is stated that 36,200 families responded in a valid matter. However, only 6549 persons were analyzed in this study. Please give more details about these figures. For example: How was the study sample drawn? How did you select the people included into the analyses?

 => The household identification survey and in-depth interview survey were conducted by the Korean Ministry of Health and Welfare. Household visit interview survey by investigators based on a structured questionnaire. Survey Population: As of 2017, general residents living in general residential facilities in 17 cities and provinces nationwidehousehold and member of the household. Sample size: About 45,000 households in 250 sampling districts (average 180 households per survey district)

=> Among the total number of households, this data was collected only by people with disabilities (6549), and no disabled participants were excluded for analysis.

-Measurement instruments, line 112: Please remove the brackets around QoL or revise the sentence otherwise.

=> we removed the brackets around QoL

-Measurement instruments, line 117: The age range of the participants was wide (0-99), the mean age being 62 years. Please give some more details about the distribution of age and the data collection. How was data collected among the youngest participants? The questions described on lines 117-152 seem to be more feasible among the adults. How about the informed consent? How was it collected from the children?

=>   age|      Freq.     Percent        Cum.

------------+-----------------------------------

          1 |          3        0.05        0.05

          2 |          1        0.02        0.06

          3 |          2        0.03        0.09

          4 |          4        0.06        0.15

          5 |          9        0.14        0.29

          6 |          8        0.12        0.41

          7 |         11        0.17        0.58

          8 |          9        0.14        0.72

          9 |          9        0.14        0.86

         10 |         14        0.21        1.07

         11 |          9        0.14        1.21

         12 |         10        0.15        1.36

         13 |          9        0.14        1.50

         14 |         14        0.21        1.71

         15 |         12        0.18        1.89

         16 |         15        0.23        2.12

         17 |         27        0.41        2.53

         18 |         15        0.23        2.76

         19 |         21        0.32        3.08

         20 |         15        0.23        3.31

         21 |         28        0.43        3.74

         22 |         24        0.37        4.11

         23 |         22        0.34        4.44

         24 |         30        0.46        4.90

         25 |         22        0.34        5.24

         26 |         17        0.26        5.50

         27 |         22        0.34        5.83

         28 |         20        0.31        6.14

         29 |         17        0.26        6.40

         30 |         17        0.26        6.66

         31 |         11        0.17        6.83

         32 |         18        0.27        7.10

         33 |         23        0.35        7.45

         34 |         26        0.40        7.85

         35 |         27        0.41        8.26

         36 |         37        0.56        8.83

         37 |         36        0.55        9.38

         38 |         36        0.55        9.93

         39 |         47        0.72       10.64

         40 |         44        0.67       11.31

         41 |         48        0.73       12.05

         42 |         45        0.69       12.73

         43 |         59        0.90       13.64

         44 |         57        0.87       14.51

         45 |         62        0.95       15.45

         46 |         65        0.99       16.45

         47 |         68        1.04       17.48

         48 |         82        1.25       18.74

         49 |         92        1.40       20.14

         50 |         99        1.51       21.65

         51 |         83        1.27       22.92

         52 |        104        1.59       24.51

         53 |         99        1.51       26.02

         54 |        102        1.56       27.58

         55 |        106        1.62       29.20

         56 |        160        2.44       31.64

         57 |        156        2.38       34.02

         58 |        154        2.35       36.37

         59 |        153        2.34       38.71

         60 |        137        2.09       40.80

         61 |        149        2.28       43.08

         62 |        173        2.64       45.72

         63 |        115        1.76       47.47

         64 |        117        1.79       49.26

         65 |        146        2.23       51.49

         66 |        148        2.26       53.75

         67 |        159        2.43       56.18

         68 |        159        2.43       58.60

         69 |        155        2.37       60.97

         70 |        178        2.72       63.69

         71 |        135        2.06       65.75

         72 |        142        2.17       67.92

         73 |        160        2.44       70.36

         74 |        192        2.93       73.29

         75 |        227        3.47       76.76

         76 |        172        2.63       79.39

         77 |        174        2.66       82.04

         78 |        179        2.73       84.78

         79 |        146        2.23       87.01

         80 |        149        2.28       89.28

         81 |        120        1.83       91.11

         82 |        105        1.60       92.72

         83 |         94        1.44       94.15

         84 |         79        1.21       95.36

         85 |         70        1.07       96.43

         86 |         54        0.82       97.25

         87 |         52        0.79       98.05

         88 |         28        0.43       98.47

         89 |         35        0.53       99.01

         90 |         18        0.27       99.28

         91 |         12        0.18       99.47

         92 |         11        0.17       99.63

         93 |          7        0.11       99.74

         94 |          8        0.12       99.86

         95 |          5        0.08       99.94

         96 |          1        0.02       99.95

         97 |          2        0.03       99.98

         99 |          1        0.02      100.00

------------+-----------------------------------

      Total |      6,549      100.00

=> The household identification survey and in-depth interview survey were conducted by the Korean Ministry of Health and Welfare. Household visit interview survey by investigators based on a structured questionnaire. Preschool children were interviewed and data collected by their parents or guardians at home.

-Table 1: It is not clear, why there is the title Quality of life in the table. It seems that the results (N, mean, %, SD) are presented according to row titles. Please specify or revise.

=> We deleted it. It is row titles.

-Table 1: The categories of marital status (yes, no) are not clear either from the table or from the lines 128-129. Please specify this.

=> We specified the categories of marital status (Yes, no) in the line of 132 with red and the table 1 as well.

-Table 1: Please indicate the unit for the amount of physical activity. On the line 152 it is stated that physical activity was assessed according to the number of minutes devoted to exercise every time. If 6549 stands for the minutes per each exercise, it sounds very high. If it covers total activity time, a time frame would be needed. Furthermore, since the variables presented in the table 1 are categorical, it might be more informative to present just the N and percentage.

=> We indicated the unit for physical activity in the line of 138 with red.

=> we revised the “physical activity was assessed according to the number of minutes devoted to exercise every time” to “physical activity was assessed by summing the number of minutes spent on each exercise.” in the line of 154 to 155 with red.

=> The categorical variables presented in the table 1 revised to present just the N and percentage.

-Table 2: Please write the DV and IV with the full words. I suppose they mark for dependent variable and independent variables, but it would be clearer to give exact words.

=> We wrote the DV and IV with the full words

-Table 2: Please indicate the adjustments in the footnote.

=> we indicated the adjustment in the footnote.

-Results: Did you consider analyzing potential interactions?

=> Analyzing potential interactions were not considered in this study.

-Discussion, line 210: It is stated that the purpose was to explore the relationship between the modes of physical activity and health-related conditions of disabled people on their health-related quality of life. However, based on the results, modes of physical activity were not analyzed, but the frequency and amount instead. Please adjust.

=> we adjusted the modes of physical activity with the frequency and amount in the line of 220 to 221 with red

-Discussion, lines 213-215: I agree with this sentence, but I did not find support for it from the results. I encourage you to replace “opportunities for physical activity” with “more frequent physical activity”.

=> We replace with it in the line of 224 with red.

-Discussion, line 227: I would encourage you to write “frequency of physical activity” (meaning number of physical activity sessions) instead of “number of activities”.

=> We replaced “number of physical activities” to “frequency of physical activity” in the line of 14, 178, 189, and 323 with red

-Discussion, line 241: In addition to “longer time” you could point out also “more frequently”.

=> We added it and revised the sentence in the line of 256 with red.

-Please be specific when writing HRQoL and when QoL. There seems to be some inconsistency throughout the text now.

=> We double-checked and revised it throughout the paper.

In-Hwan Oh, M.D., Ph. D

Department of Preventive medicine

School of Medicine

26 Kyungheedae-ro, Dongdaemun-gu, Seoul,

Republic of Korea, 02447

Tel: 82-2-961-2304, Fax: 82-2-969-0792

E-mail: parenchyme@gmail.com
